# Beyond Macromolecules: Extracellular Vesicles as Regulators of Inflammatory Diseases

**DOI:** 10.3390/cells12151963

**Published:** 2023-07-29

**Authors:** Kaushik Das, Subhojit Paul, Tanmoy Mukherjee, Arnab Ghosh, Anshul Sharma, Prem Shankar, Saurabh Gupta, Shiva Keshava, Deepak Parashar

**Affiliations:** 1Department of Cellular and Molecular Biology, The University of Texas at Tyler Health Science Center, Tyler, TX 75708, USA; 2School of Biological Sciences, Indian Association for the Cultivation of Science, Jadavpur, Kolkata 700032, India; bcsp3@iacs.res.in (S.P.); bcag3@iacs.res.in (A.G.); 3School of Medicine, The University of Texas at Tyler Health Science Center, Tyler, TX 75708, USA; tanmoy.mukherjee@uttyler.edu; 4Department of Molecular, Cell and Cancer Biology, University of Massachusetts Medical School, Worcester, MA 01605, USA; anshul.mic@gmail.com; 5Department of Neurobiology, The University of Texas Medical Branch, 301 University Blvd, Galveston, TX 77555, USA; premshankar.1506@gmail.com; 6Department of Biotechnology, GLA University, Mathura 281406, India; saurabh.gupta@gla.ac.in; 7Department of Medicine, Division of Hematology & Oncology, Medical College of Wisconsin, Milwaukee, WI 53226, USA

**Keywords:** extracellular vesicles, classification, inflammation, inflammatory diseases, coagulation, therapeutics

## Abstract

Inflammation is the defense mechanism of the immune system against harmful stimuli such as pathogens, toxic compounds, damaged cells, radiation, etc., and is characterized by tissue redness, swelling, heat generation, pain, and loss of tissue functions. Inflammation is essential in the recruitment of immune cells at the site of infection, which not only aids in the elimination of the cause, but also initiates the healing process. However, prolonged inflammation often brings about several chronic inflammatory disorders; hence, a balance between the pro- and anti-inflammatory responses is essential in order to eliminate the cause while producing the least damage to the host. A growing body of evidence indicates that extracellular vesicles (EVs) play a major role in cell–cell communication via the transfer of bioactive molecules in the form of proteins, lipids, DNA, RNAs, miRNAs, etc., between the cells. The present review provides a brief classification of the EVs followed by a detailed description of how EVs contribute to the pathogenesis of various inflammation-associated diseases and their implications as a therapeutic measure. The latter part of the review also highlights how EVs act as a bridging entity in blood coagulation disorders and associated inflammation. The findings illustrated in the present review may open a new therapeutic window to target EV-associated inflammatory responses, thereby minimizing the negative outcomes.

## 1. Introduction

The ability of cells to communicate with each other holds an important step in the differentiation and development of multicellular organisms. Numerous mechanisms govern how cells interact with each other, such as cellular secreted molecules, direct interaction between the adjacent cells through the cell-adhesion molecules, and the formation of cytoplasmic bridges or nanotubules [1]. However, a growing body of evidence identifies a unique mechanism by which cells convey signals between one another, the release of extracellular vesicles (EVs) [2,3,4]. EVs are membrane-enclosed nano-sized bodies, shown to be released from almost every cell type [5,6]. As EVs are derived from cells, they often carry cellular components such as proteins, lipids, and genetic materials in the form of DNA, RNA, microRNA (miRNA), etc. [7], and upon transferring these bioactive molecules, EVs generally modulate the function of the target recipient cells [8,9,10]. A wide variety of non-coding RNAs (ncRNAs) including miRNAs regulate the fundamental cellular processes which can be therapeutically targeted in the context of cancer [11,12]. The uptake mechanisms of EVs by the recipient cells include the direct fusion of EVs with the plasma membrane or endocytosis [4,8,13]. EVs are readily detected in every biological fluids including blood, urine, saliva, synovial fluid, sputum, breast milk, bronchoalveolar lavage fluid (BALF), and cerebrospinal fluid (CSF) and even in interstitial spaces between the cells [6,14,15,16,17,18]. Based on the biogenesis, content, size, and function, EVs are extensively categorized into three major groups, microvesicles, exosomes, and apoptotic bodies (Figure 1) [5,6].

*Microvesicles*. Microvesicles (MVs) or microparticles (MPs) or ectosomes are recognized as plasma membrane ‘buds’ of the cells [7,19]. The crosstalk among cytoskeletal components such as actin and microtubules, molecular motor proteins such as kinesin and myosin, fusion machineries such as soluble N-ethylmaleimide-sensitive factor activating protein receptor (SNARE), and tethering factors essentially regulates the formation and release of MVs from the cells [2,3,20,21,22,23]. The size of MVs is believed to range from 100 nm to 1 µm in diameter [5,6,14]. MVs, because of being generated by plasma membrane outward budding, are shown to carry cytosolic and plasma membrane-associated proteins such as tetraspanins, which often serve as a universal marker for the MVs, regardless of the cells’ origin [24]. Moreover, cytoskeletal proteins such as heat shock proteins, integrins, and proteins associated with posttranslational modifications including glycosylation, phosphorylation, etc., are, sometimes, found to be enriched in MVs [25].

*Exosomes*. Exosomes, the smaller EV class having a diameter of 30–150 nm [26], are generated by the endocytic mechanism [25]. Typically, invagination of the early endosomal membrane produces these exosomes which are matured into multivesicular bodies (MVBs) [25]. MVBs are eventually fused with the plasma membrane, thereby releasing the exosomes outside the cells [25]. Exosomes’ biogenesis often requires the active involvement of endosomal sorting complexes required for the transport (ESCRT) pathway [25]; therefore, ESCRT pathway-associated molecules including TSG101, Alix, HSP90β, and HSC70 are shown to be present in the exosomes [27,28], which are also used as exosomal markers. However, ESCRT-independent exosomal biogenesis also occurs, which is reported to be associated with sphingolipid ceramide [29].

Recently, a unique exosomal release mechanism has been identified which involves the autophagic pathway. Autophagy is the process of eliminating non-functional and futile components of the cells depending on lysosomal mechanisms [30]. The sequestration of a cytoplasmic portion by a membranous organelle, called a phagophore, generates autophagosomes, which in turn fuse with the MVBs to produce the amphisomes [31,32]. Amphisomes are often found to be enriched with endosomes as well as autophagosome markers, LC3 and CD63, respectively. Moreover, cytosolic DNA and nucleosomes are also present in the amphisomes. Amphisomes are either fused with the plasma membrane, resulting in the release of amphisomal content including the exosomes outside the cell, a phenomenon called ‘exophagy’, or their fusion with the lysosomes leads to the degradation of the amphisomal components by lysosomal enzymes. 

*Apoptotic bodies*. In contrast to MVs and exosomes, apoptotic bodies are larger in size, ~50 nm to 5 µm in diameter [33]. These are released from the apoptotic cells via the separation of the plasma membrane from the cytoskeleton due to immense hydrostatic pressure, generated during the cell contraction [34]. Apoptotic bodies are often found to contain cell organelles, nuclear chromatin, and a few glycosylated proteins; therefore, mitochondrial proteins, such as HSP60, Golgi, and endoplasmic reticular proteins, such as GRP78, and nuclear histones appear to be markers for apoptotic bodies [33,35,36,37]. A basic comparison among different classes of EVs is shown in Table 1.

EV isolation procedures: A comparative analysis. The present section briefly discusses different techniques of EVs isolation in a comparative approach. Currently, the widely accepted procedures for the isolation of EVs include centrifugation, precipitation, size exclusion, affinity purification, and micro-/nano-fluidics or chips [38]. Table 2 briefly summarizes the purity, yield, time consumption, and sample volume required for the isolation of EVs by different procedures in a comparative manner [38,39].

*Centrifugation*. This is the most commonly used method for isolating EVs by several research groups, principally based on the particle size, density, shape, and viscosity of the medium. This is further classified into differential ultracentrifugation, density-gradient centrifugation, and rate-zonal centrifugation [39,40]. (1) Differential centrifugation separates the EVs based on the size, shape, and density [39,41]. The influencing factors in this method include temperature, sample dilution, and duration of centrifugation [42,43]. Although the procedure is easy, has average yield, and needs no additional steps for the preparation of samples, it is time-consuming, laborious [39,44,45,46], and incapable of differentiating between different EVs types [47]. In addition, protein contaminants are the major issue in this EV isolation procedure [46]. (2) In contrast to differential centrifugation, density-gradient centrifugation employs a preconstructed density-gradient medium such as sucrose and iodixanol for the isolation of EVs [39,48]. This method has the advantage of separating EVs from the contaminating proteins [39], and different types of EVs can be separated according to their density [49]. However, average yield and the need for longer isolation time are the two major pitfalls of this approach [50,51]. (3) Rate-zonal centrifugation, on the other hand, utilizes the combined principle of density-gradient and sedimentation in which the sample is loaded on top of the tube, and following centrifugation, EVs with higher density are shown to pass through the dense layer as compared to lighter EVs [38]. The additional advantages of this technique over the other centrifugation procedures are that EVs with same density but different size can be separated [52] and the high yield recovery of the EVs [39].

*Precipitation*. This method employs the use of a water-excluding compound, such as Polyethylene glycol (PEG), which is mixed with the EV sample, followed by centrifugation or filtration. PEG dries up the sample, leading to the precipitation of the other molecules [53,54,55]. Although this method is easy and applicable for both small and larger volume of samples, more often it results in the co-precipitation of the non-EV components. Therefore, precipitation is always combined with other techniques to improve the quality and selectivity [39,54,56,57].

*Size exclusion*. This procedure explores the different size distributions of EVs for their isolation. Size exclusion techniques include ultrafiltration, sequential filtration, isolation kits, field-flow fractionation, size-exclusion chromatography, and hydrostatic filtration dialysis. (1) In ultrafiltration, the EVs samples pass through different pore-sized membrane filters, leading to the separation of the EVs based on their size and molecular weight [39,54]. Despite the fast and inexpensive separation of the EVs [39,56], this method has several disadvantages. Often, the EVs become entrapped in the membrane [39,56]. Moreover, poor efficiency and EVs’ deformation due to membrane pressure further lead to the lower efficiency of the process [39,56,58,59]. (2) Sequential filtration, a semi-automated technique, is basically a system composed of multiple filters of different sizes. When an EV sample is loaded, the larger particles are trapped in the filters, and the smaller ones pass through. Although this technique is fast [58], it often results in membrane plugging and hence low yield [58,60,61]. (3) Recently, isolation kits have been developed which also separate EVs based on their size. For example, Exomir Kits are composed of two membranes: the upper one is of a higher pore size (200 nm), whereas the bottom one has a lower pore size (20 nm) [39]. Another isolation kit, ExoTIC, contains multiple filters, and the EV samples, when applied to it, are separated according to their size. These kits often produce high yield EVs [62]. (4) In field-flow fractionation, the EV samples are loaded into a chamber in which a crossflow is generated. The larger particles, due to the cross-flow, are positioned on the chamber wall, whereas the smaller particles are eluted first [39,63]. This technique is fast, is efficient, provides higher recovery, and facilitates the isolation of EVs from a very small sample volume [64]. (5) Size-exclusion chromatography allows the elution of larger particles from the column followed by the release of smaller particles through the pores [39,58,65,66]. This not only obtains the biological integrity of the EVs but also offers no damage of sample pre-treatment [58,65]. (6) Hydrostatic filtration dialysis employs hydrostatic pressure for the isolation of EVs. It is a tube-based technique in which the small particles are diffused through the membrane, whereas the larger ones are retained in the tube [39,67].

*Affinity purification*. Affinity purification of EVs involves antibody-mediated purification of the EVs against surface antigens [39]. In this technique, the purity of the EVs is shown to be the highest [39]; however, at the same time, poor yield limits the efficiency of the method [39,57]. Also, the availability of antibodies against unique antigens on the EVs further adds to the difficulties of affinity purification [38]. However, combinational techniques, in association with affinity purification, are found to be quite effective [68].

*Micro/nano fluidics or chips*. Biochemical features such as electrophoretic, acoustic, and electromagnetic properties of the EVs are often explored to develop micro-/nano-chips for the isolation of EVs [39,54]. For example, the development of micro-chips is based on the size, immunoaffinity, and density of the EVs [38]. Nanowires, viscoelastic flow, and nano-sized deterministic lateral displacement (nano-DLD) are the other techniques that fall into this category. The nanowires’ principle is very similar to size-exclusion chromatography, which contain silicon micropores [38]. The elastic lift forces of different sized EVs vary in a viscoelastic medium, which is utilized in EV isolation by the viscoelastic flow [69,70]. On the other hand, nano-DLD utilizes the pillar-array-based microfluidic mechanism for the isolation and analysis of the EVs [69]. The acoustic separation method employs the ultrasonic radiation, in which the EVs are exposed, for the separation of the EVs. Based on their size, the frequency of the waves is controlled to separate the EVs. The larger particles, influenced by the heavier waves, move to the pressure node at a faster rate [39,71]. This often leads to the yield of highly purified EVs [72].

**Table 2 cells-12-01963-t002:** Purity, yield, time consumption, and sample volume required for different EVs’ isolation procedures in a comparative approach [38,39].

Isolation Method	Purity	Yield	Time Consumption	Sample Volume Needed
*Centrifugation*				
Differential ultracentrifugation	Low	Low-moderate	8 h	100 mL
Density-gradient centrifugation	>Differential ultracentrifugation	Low-moderate	20 h	1 mL
Rate-zonal Centrifugation	High	>Density-gradient centrifugation	2 h	0.5 mL
*Precipitation*	Low		16 h	1 mL
*Size exclusion*				
Ultrafiltration	>Differential ultracentrifugation	Very high	18 h	0.5 mL
Sequential filtration	High	<Differential Ultracentrifugation	-	150 mL
Isolation kits	High	High	-	10–100 µL
Field-flow fractionation	High	High	<1 h	100 µL
Size-exclusion Chromatography	High	High	~1.5 h	50 mL
Hydrostatic Filtration dialysis	-	>Differential ultracentrifugation	9 h	15–200 mL
*Affinity purification*	Very high	Poor	~45 min	~100 µL
*Micro/nano-fluidics or chips*				
Immune microfluidic	-	Almost 100%	~100 min	30 µL
Viscoelastic flow	Very high	Very high	5–25 min	<100 µL
Acoustic separation	Very high	Very high	25 min	100 µL

Heterogeneity in EV preparations: Minimal Information for Studies of Extracellular Vesicles 2018 (MISEV2018). In the past three decades, the advancement of EV research has also increased the complexity of EV characterization. Depending on the cells of origin, biogenetic mechanisms, and various physiological and pathological functions, different research groups apply different terminology for the EVs, such as exosomes, microparticles, microvesicles, ectosomes, apoptotic bodies, oncosomes, and many others. However, the disparity in size within different methods of EV preparation often turns out to be the primary limitation for EV characterization. In this regard, the International Society for Extracellular Vesicles (ISEV) proposed a guideline for the isolation and characterization of EVs, termed as ‘Minimal Information for Studies of Extracellular Vesicles’ (MISEV), in 2014 which was further updated in 2018 [73]. A worldwide ISEV survey from 2015 [74] indicates that the differential ultracentrifugation was the most frequently used technique for separating and concentrating EVs over the other conventional methods, such as density-gradient centrifugation, precipitation, filtration, size-exclusion chromatography, affinity purification, etc., with moderate purity and recovery. However, for better specificity and recovery, several other techniques were further used which are mentioned in MISEV2018 [73]. These include tangential flow filtration and variations thereon, asymmetric flow field-flow fractionation, field-flow fractionation, field-free viscoelastic flow, variations on size exclusion chromatography (SEC), acoustics, alternating current electrophoretics, ion exchange chromatography, fluorescence-activated sorting, microfiltration, DLD arrays, novel precipitation/combination techniques, novel immunoisolation or other affinity isolation technologies, hydrostatic filtration dialysis, a wide variety of microfluidics devices which combine one or more principles, as mentioned above, and high-throughput/high-pressure methods including fast protein liquid chromatography/high performance liquid chromatography (FPLC/HPLC) involving some chromatography techniques [73]. Table 3 briefly describes the differences in purity and recovery among various EV isolation procedures in accordance with MISEV2018 [73].

Selectivity of EVs in the uptake by target cells. There is mixed evidence which indicates the movement of EVs towards specific target cells. Although EVs are shown to be non-selectively taken up by a wide variety of recipient cells [75], at times, the release of specific morphogens by the target recipient cells may guide the EVs towards them [76]. However, an interesting study by Sharif et al. demonstrates that Wharton’s jelly-mesenchymal stem cell (WJ-MSC)-derived EVs specifically deliver miR-124 to glioblastoma multiforme (GBM), resulting in the down-regulation of GBM migration while increasing its chemosensitivity [77]. This indicates the possibility of a ligand–receptor interaction in the specific uptake of EVs by the target recipient cells. In this context, the role of EVs’ membrane proteins, lipids, and glycans becomes indispensable. Table 4 briefly summarizes the role of EVs’ membrane components in the uptake of EVs by target recipient cells.

*EVs’ membrane proteins*. EVs’ membrane proteins play a major role in their uptake by specific target cells. Tetraspanins (CD63, CD9, CD82, and CD81), the abundantly expressed molecules on the surface of EVs [78], in association with other adhesion molecules such as intercellular adhesion molecule (ICAM) [79] essentially mediate the docking and uptake of EVs by the recipient cells upon interacting with cellular integrins and other adhesion molecules [78]. Hoshino et al. further demonstrate that α6β4- and α6β1-integrin + EVs are associated with lung metastasis, whereas αvβ5-integrin + EVs are involved in liver metastasis, and targeting the EVs’ integrins not only interferes with the EVs’ uptake but also decreases the EV-associated metastasis [80]. 

*Lipids of EVs’ membrane*. EVs are enriched with a negatively charged phospholipid, phosphatidylserine (PS), which is indirectly identified by the growth arrest-specific protein 6, Gas6, leading to the activation of Mer receptor tyrosine kinase (MERTK) on the surface of macrophages, thereby facilitating the EVs’ uptake and associated anti-inflammatory response [81].

*EVs’ membrane glycans*. In most cases, glycans are abundantly found on the surface of the EVs, and targeting glycans, more specifically proteoglycans, is believed to reduce EVs’ uptake by interfering with the glycans–lectin interaction [82]. Moreover, mannose-containing glycoproteins are glycan structures that are often found on the EVs’ membrane whose inhibition significantly down-regulates the uptake of the EVs by ovarian cancer cells [36].

**Table 4 cells-12-01963-t004:** The role of EVs’ membrane components in the uptake of EVs by target recipient cells.

Membrane Component	Type	Specific Name	Function	Reference
Membrane proteins	Tetraspanins	CD63, CD9, CD82, CD81	EVs’ tetraspanins, in association with adhesion molecules, such as ICAM, bind to cellular integrins and other adhesion molecules, hence promoting EVs uptake by target recipient cells	[78]
	Integrins	α6β4, α6β1, αvβ5	Targeting α6β4- and α6β1-integrins on the EVs decreases lung metastasis, whereas αvβ5-integrin targeting of EVs reduces liver metastasis, via interfering with the uptake of the EVs	[80]
Membrane lipids	Glycero-Phospholipids	PS	EVs-PS is indirectly recognized by Gas6, leading to MERTK activation in the recipient macrophages, thereby facilitating EVs’ uptake and associated anti-inflammatory response	[81]
Membrane glycans	Proteoglycans	-	Proteoglycans are abundant on the EVs’ surface, and targeting proteoglycans would reduce EVs’ uptake by inhibiting the glycan–lectin interaction	[82]
	Mannose-containing glycoproteins	-	Mannose-containing glycoproteins are enriched on the EVs’ surface, the blocking of which significantly attenuates EVs’ uptake by ovarian cancer cells	[36]

*Abbreviations:* CD, cluster of differentiation; ICAM, intercellular adhesion molecule; PS, phosphatidylserine; Gas6, growth arrest-specific protein 6; MERTK, Mer receptor tyrosine kinase.

EVs in various diseases. The abundance and heterogeneity of different cargoes entrapped within EVs often turn out to be important biomarkers in various pathophysiological conditions. For example, the level of pro-coagulant tissue factor (TF) expression is shown to be well-elevated on the plasma EVs of Gram-negative sepsis-induced urinary tract infection (UTI) patients, which often contributes significantly to the hyper-coagulative responses [83]. In contrast, EVs derived from activated platelets are believed to confer anti-coagulative effects [84]. In the case of atherosclerosis, the plaque-derived EVs transport ICAM-1 to the endothelial cells depending on the PS, thereby leading to the recruitment of inflammatory cells to promote atherosclerotic plaque progression [85]. Moreover, in acute kidney injury (AKI), fetuin-A and AQP1 + EVs may be used as diagnostic biomarkers. The level of fetuin-A is significantly up-regulated in the urinary EVs, whereas EVs’ AQP1 expression is shown to be down-regulated in AKI [86]. Furthermore, ten signature miRNA molecules (miR-199a-5p, miR-143-3p, miR-4532, miR-193b-3p, miR-199b-3p, miR-199a-3p, miR-629-5p, miR-25-3p, miR-4745-3p, and miR-6087) are found to be up-regulated, whereas another ten miRNAs (miR-23b-3p, miR-10a-5p, miR-141-3p, miR-98-5p, miR-382-5p, miR-200a-3p, miR-200c-3p, miR-483-5p, miR-483-3p, and miR-3911) are significantly down-regulated in the human follicular fluid (HFF)-derived EVs of polycystic ovary syndrome (PCOS) patients, which can serve as PCOS biomarkers [87]. The cerebrospinal fluid (CSF) of patients with Parkinson’s disease (PD) is shown to be enriched with α-synuclein + EVs which facilitate the aggregation of α-synuclein in healthy cells, leading to the progression of PD [88]. Circulating EVs from the differentiating myoblasts actively participate in the enhancement of muscle regeneration during congenital myopathies, and thus the elevated level of circulating EVs could be considered as the biomarker for congenital myopathy progression [89]. The composition of microbial EVs in the feces, blood, and urine of patients with gastrointestinal tract disease is significantly altered as compared to healthy individuals, and hence these EVs have the potential of being recognized as a diagnostic biomarker for microbial infection [90]. Numerous studies have mentioned the important contributions of EVs in the progression of cancer. For example, breast cancer cell-derived EVs transfer miR-125b to the normal fibroblasts in the tumor microenvironment (TME) rendering their transformation into cancer-associated fibroblasts (CAFs) [91]. Moreover, the population of triple-negative breast cancer (TNBC) cell-secreted EVs is shown to be significantly increased in the presence of FVIIa, which imparts epithelial to mesenchymal transition (EMT) to the EMT-negative cells via miR-221 transfer, leading to the progression of TNBC [9]. Again, the level of miR-144 is shown to be well-elevated in the EVs derived from nasopharyngeal carcinoma (NPC) which is readily transferred to the endothelial cells following EVs uptake, thereby resulting in the enhanced migration, invasion, and angiogenesis of the endothelial cells [92]. In the majority of instances, EVs have been associated with the modulation of inflammatory responses in different ways and are often considered an important regulator in various inflammation-associated diseases.

The role of EVs in inflammatory diseases. Inflammation, the defense mechanism of the immune system against harmful stimuli [93] such as radiation [94], toxic compounds [95], damaged cells [96,97,98,99], and most importantly pathogens [100], is characterized by tissue redness, swelling, heat, pain, loss of tissue functions, and recruitment of the immune cells at the site of infection [101,102,103], the results of which help eliminating the harmful cause and initiate the healing process [104,105,106]. However, just as ‘too much of anything is bad’, prolonged inflammation often gives rise to several chronic disorders [107,108,109,110]. Therefore, a balance between pro- and anti-inflammatory responses is a prerequisite in the removal of injurious stimuli with minimal damage to the host. Inflammation is often shown to play a pivotal role in various pathophysiological anomalies such as neurological disorders, cardiovascular diseases, respiratory syndrome, defects in the digestive and integumentary systems, disease associated with musculoskeletal, urinary, and reproductive systems, and endocrine as well as lymphatic disorders. The emerging role of EV-associated inflammatory responses in various diseased conditions is briefly illustrated in the present review (Table 5). Moreover, it has been established that inflammation and blood coagulation are intrinsically related: the activation of one process often leads to the activation of the other [111,112,113]. The latter part of the review focuses on how EVs influence the inflammatory responses in various coagulation-associated disorders.

*Evs in neuroinflammatory diseases*. Emerging evidence implicates the active involvement of Evs in various neuroinflammatory diseases. For example, the concentration of plasma Evs is shown to be significantly up regulated in the central nervous system (CNS). Autoimmune disease, multiple sclerosis (MS) [114], and EVs of endothelial as well as platelet origin from the plasma of MS patients have been revealed to induce blood–brain barrier (BBB) permeability, leading to the transmigration of myeloid- and T-cells into the CNS, thereby contributing to the neuropathology in MS [115,116,117]. Moreover, EVs in the plasma and CSF of patients suffering from neurodegenerative disorders such as Alzheimer’s disease (AD), PD, etc., are enriched with neurotoxic molecules including β-amyloid (Aβ), α-synuclein, and tau, whose origin are believed to be microglia and neuronal cells, and the uptake of toxic molecule-laden EVs to the local and distant neurons contributes to the neuronal loss, the characteristic feature of neurodegenerative disorders [118,119,120,121]. Another neurodegenerative disease, Creutzfeldt–Jakob disease (CJD), is caused by the misfolded and transmissible form of the prion protein (PrP) PrP^Sc^. PrP^Sc^ is readily detected in the plasma EVs of CJD patients [122], and the selective packaging of PrP^Sc^ into the neuronal EVs often contributes to the EV-associated pathogenetic spread of CJD [123]. EVs often contribute to CNS infection. For example, JC polyomavirus (JCPyV), the causative agent of progressive multifocal leukoencephalopathy (PML), is shown to be transferred via serum EVs between glial cells and is highly infectious and leads to the pathogenesis of PML [124]. Furthermore, *Plasmodium*-infected red blood cells and other host cells have been demonstrated to release a significant amount of EVs in circulation [125] which contribute to the pathogenesis of cerebral malaria (CM), the most severe form of malaria, and targeting EV biogenesis has proven to be highly effective against CM in an animal model system [126]. In contrast to the above, EVs have also proved to be beneficial in a few instances.

In stroke, MSC-derived EVs have been reported to perturb the microglial differentiation of pro-inflammatory M1 phenotypes, thereby prohibiting neuroinflammation and brain injury following middle cerebral artery occlusion (MCAO) in rats [127]. Again, during spinal cord injury (SCI), infiltrating macrophages release NADPH oxidase 2 (NOX2)-loaded EVs which are readily taken up by the injured neuronal axons, and inside the neurons, NOX2 inactivates PTEN, thereby stimulating the PI3K-AKT pathway to regenerate neuronal outgrowth [128]. In addition, microglial EVs are shown to be enriched with miR-124-3p in conditions such as traumatic brain injury (TBI), which not only inhibits neuronal inflammation but also induces neurite outgrowth via PDE4B-targeted down-regulation of the mTOR signaling pathway [129]. In the majority of the above-mentioned studies, differential centrifugation techniques have been employed to isolate the EVs, which often reduces the purity, always leaving behind the possibilities of protein contaminants’ presence in the EV preparation which could affect the inflammatory responses of the EVs. However, Asai et al. [118] and Robertson et al. [122] used ultracentrifugation followed by density-gradient centrifugation for isolating the EVs, which improves the purity of the EVs markedly. In addition to this, Guo et al. utilized ExoQuick-TC PLUS followed by ultracentrifugation for EVs isolation which also yields highly purified EVs [121]. Figure 2 illustrates how EVs contribute to the progression of different neuroinflammatory diseases via different mechanisms. 

*EVs and cardiovascular inflammatory responses*. Inflammation plays a key role in the pathogenesis of various cardiovascular diseases such as atherosclerosis, myocardial infarction and ischemic heart disease, heart failure, aneurysms, etc. 

A growing body of evidence highlights the active participation of EVs in these inflammation-associated cardiovascular anomalies. For example, during initial atherogenic stages, EVs from atherogenic plaque, circulating monocytes, and neutrophils induce the endothelial expression of ICAM-1. This facilitates leukocyte recruitment, adhesion, and trans-endothelial migration, mostly via the activation of pro-inflammatory signaling pathways [85,130,131]. This is followed by the plaque maturation stages, wherein EVs from platelets and adipose cells play a pivotal role by enhancing the formation of foam cells depending on the pro-inflammatory signaling. Platelet-derived EVs trigger the macrophages’ phagocytosis of oxidized LDL (ox-LDL) [132]. Adipose cell-derived EVs, on the other hand, perturb the cholesterol efflux of macrophages [133]. Both the platelet- and adipose cell-derived EVs are shown to stimulate the formation of foam cells. In the final stage, atherosclerotic plaque progression essentially requires calcification, and EVs from pro-inflammatory macrophages are shown to induce microcalcification both in human and murine systems [134,135]. EVs also play a pivotal role in inflammation-associated myocardial infarction (MI) and ischemic heart disease. For example, EVs in the myocardium, originating from cardiomyocytes and endothelial cells, trigger the secretion of pro-inflammatory cytokines and chemokines from infiltrating monocytes. These pro-inflammatory molecules contribute to the pathogenesis of MI and ischemic heart disease [136]. On the other hand, EVs’ miR-155 is reported to be transferred from activated macrophages to cardiac fibroblasts. This leads to the inhibition of fibroblast proliferation and triggers the inflammatory responses, thereby contributing to the cardiac rupture [137]. EVs are also demonstrated to be involved in heart failure (HF)-associated inflammatory responses. For example, cardiac fibroblasts are well known for releasing miR-27a*- and miR-21*-laden EVs, capable enough of promoting cardiac hypertrophy [138,139]. On the other hand, cardiac hypertrophy is also driven by cardiomyocytes which promote fibroblast proliferation via the release of miR-217-laden EVs [140]. Moreover, the role of EVs in aneurysm is widely documented. For example, neutrophil EVs, in the intraluminal thrombus of aortic aneurysms, are known for carrying ADAM10 and ADAM17 which, due to their proteolytic activities, cause the degradation of aortic walls [141]. Additionally, ficolin-3 + platelet-derived EVs are well elevated in the plasma of aortic aneurysms patients which contribute to the progression of aneurysms [142]. In the above-mentioned studies, the authors used either differential centrifugation or ultracentrifugation for EVs’ isolation, which reduces EVs’ purity and hence could influence the inflammatory behavior of the EVs. Figure 3 briefly summarizes the role of EVs in the progression of different cardiovascular inflammatory diseases.

*EVs in respiratory inflammatory diseases*. EVs often influence inflammation-associated respiratory diseases, such as acute lung injury (ALI) and acute respiratory distress syndrome (ARDS), chronic obstructive pulmonary disease (COPD), pulmonary hypertension (PH), idiopathic lung fibrosis (ILF), asthma, etc. In ALI and ARDS, EVs are released into the BALF upon infection (LPS or Gram-negative bacteria) or sterile stimuli (acid aspiration or oxidative stress) from alveolar macrophages or alveolar type-I epithelial cells, respectively. These EVs trigger the release of pro-inflammatory cytokines and mediators from naïve alveolar macrophages, leading to the development of lung inflammation [143]. In the case of COPD, bronchial epithelial cell-derived EVs are shown to be enriched with miR-210. These miR-210-laden EVs are associated with autophagy functions and myofibroblasts differentiation, the dysregulation of which leads to the pathogenesis of COPD [144]. Furthermore, in PH, more specifically pulmonary arterial hypertension (PAH), miR-143-laden EVs from pulmonary arterial smooth muscle cells (PASMCs) promote migration and angiogenesis of pulmonary arterial endothelial cells (PAECs) [145]. These contribute to the pathogenesis of PH. BALF-EVs of ILF patients have an abundance of WNT5A, believed to originate from the lung fibroblasts, and are shown to promote fibroblast proliferation and the pathology of ILF [146]. In asthma, plasma EV-associated miR-145 plays a crucial role in epithelial and smooth muscle cell functions [147] related to inflammation. The inhibition of miR-145 is often observed during asthma which is accompanied by low eosinophilic inflammation, Th2 cytokine production, airway hyperresponsiveness, and hypersecretion of mucous, characteristic features of asthma-induced bronchial stress [148]. The use of ultracentrifugation in EV isolation, in the above-mentioned studies, limits the purity of the EVs except for Martin-Medina et al. who used highly purified EVs, isolated by ExoQuick followed by ultracentrifugation, in their study [146]. Figure 4 briefly demonstrates how EVs play their part in various respiratory inflammatory syndromes.

*EVs in inflammatory diseases of the digestive system*. A growing body of evidence indicates that EVs also play important roles in the inflammatory diseases of the digestive system, such as necrotizing enterocolitis (NEC) and inflammatory bowel disease (IBD). Numerous studies have demonstrated the active involvement of EVs in influencing NEC and IBD; however, the present review highlights a few of them. NEC is considered to be one of the catastrophic diseases of newborns with mortality rates of ~20–30% [149]. EVs from stem cells often show protective responses against NEC, indicating the therapeutic potential of the stem cell-derived EVs in NEC. Pisano et al., in a recent study, demonstrated that pre-treatment of intestinal epithelial cells (IEC) with bone marrow (BM)-derived EVs, which are abundant in the breast milk, rescues IEC against hypoxia/reoxygenation (H/R)-triggered inhibition of proliferation and induction of apoptosis in a rat model [149]. Furthermore, amniotic fluid stem cell (AFSM)-derived EVs are shown to promote epithelial proliferation and anti-inflammation, leading to the regeneration of normal intestinal epithelium, ultimately contributing to the intestinal recovery following NEC [150]. IBD, the other inflammatory disease of the gastrointestinal tract, is caused by the dysbiosis of the intestinal microenvironment, currently affecting more than 3.5 million people worldwide [151]. IECs, under physiological conditions, produce TGF-β1-laden EVs which induce regulatory T-cells (T_reg_) and immunosuppressive dendritic cells, thereby decreasing the severity of IBD [152]. Moreover, mast cell (MC)-derived EVs transfer miR-223 to the IECs, hence targeting IECs’ Claudin 8 (CLDN8), resulting in the loss of intestinal epithelial tight junctions which leads to increased intestinal epithelial permeability, the characteristic feature of IBD [153]. IBD-induced injury to the epithelial barrier triggers the release of annexin A1 (ANXA1) + EVs from the IECs, which is associated with the activation of the wound repair process [154]. Unlike others, Li et al. [150] and Jiang et al. [152] used the ExoQuick kit for the isolation of EVs in their studies, which not only improves the yield as compared to conventional ultracentrifugation but also consumes less time. However, ExoQuick-purified EVs without subsequent centrifugation steps may result in a high degree of lipoprotein contamination.

*The role of EVs in integumentary inflammatory diseases*. EVs are sometimes shown to be involved in the inflammatory responses of various integumentary diseases such as systemic lupus erythematosus (SLE), psoriasis, atopic dermatitis (AD), etc. In SLE, the number of circulating EVs is found to be well-elevated, and those EVs target the endothelial cells leading to the secretion of pro-inflammatory cytokines, induction of endothelial apoptosis, and enhancement of vascular permeability, ultimately contributing to secondary tissue leukocyte infiltration [155]. In psoriasis, interferon α (IFN-α)-induced mast cell-derived EVs transfer cytoplasmic phospholipase A_2_ (PLA_2_) to nearby CD1a-expressing cells, thereby generating neo lipid antigens and their recognition by CD1a-reactive T-cells to induce the release of IL-22 and IL-17A, ultimately leading to skin inflammation [156]. Furthermore, in AD patients, *Staphylococcus aureus*-derived EVs (SEVs) trigger dermal microvascular endothelial cells (DMECs) to induce the expression of E-selectin, ICAM-1, VCAM-1, and IL-6 release via TLR4-NF-ĸB signaling, thereby promoting leukocytes’ adhesion to the endothelium and their subsequent transmigration to promote AD progression [157]. In the above-mentioned studies, the isolation of EVs was carried out through differential or ultracentrifugation. However, it is important to note that these methods leave behind the possibility of soluble protein contaminants, which can have a significant impact on the inflammatory responses under investigation. 

*EVs’ role in musculoskeletal inflammatory diseases*. An increasing body of evidence indicates that EVs also play a crucial role in inflammatory responses associated with musculoskeletal diseases, which include osteoporosis (OP), osteoarthritis (OA), etc. For example, oxidative stress and aging result in the elevated expression of miR-183-5p in the EVs isolated from bone marrow interstitial fluid (BMIF). miR-183-5p is shown to arrive from aged bone marrow stromal cells (aBMSCs) and is capable of targeting heme oxygenase-1 (Hmox1) in young BMSCs (yBMSCs), thereby not only inhibiting the proliferation and osteogenic differentiation of yBMSCs but also promoting yBMSCs senescence, the characteristic features of OP [158]. In OA, EVs from IL-1β-stimulated synovial fibroblasts (SFBs) are observed to induce MMP-13 and ADAMTS-5, whereas inhibiting COL2A1 and ACAN expression in articular chondrocytes contributes to the pathogenesis of OA [159]. Unlike others, Kato et al. [159] used both ultracentrifugation and ExoQuick for the isolation of EVs in their studies. As stated before, the use of ExoQuick without subsequent ultracentrifugation improves the yield significantly but leaves behind the possibility of lipoprotein contamination. Figure 5 briefly illustrates the role of EVs in different inflammation-associated diseases of the digestive system, integumentary system, and musculoskeletal system.

*The role of EVs in urinary inflammatory diseases*. EVs also play critical roles in the progression of several urinary inflammatory diseases. For example, the level of plasma or urine-derived EVs is often used as a predictive biomarker for the progression of acute kidney injury (AKI) [160]. Guan et al. showed that hypoxia or ischemia-reperfusion (I/R)-induced injured tubular epithelial cells (TECs) release a significant amount of miR-150-laden EVs which develop profibrotic manifestations to renal fibroblasts. Moreover, the expression of urinary EVs’ chemokine (C-C motif) ligand 2 (CCL2) mRNA is shown to be significantly higher in IgA nephropathy (IgAN) patients as compared to other glomerulopathy controls, which is correlated with the tubular interstitial inflammation and C3 deposition, reflecting renal injury and impaired renal functions [161]. Again, as in the majority of cases, using ultracentrifugation to isolate EVs frequently results in a drop in EV purity.

*EVs’ role in inflammatory diseases of the reproductive system*. In the uterine microenvironment (UME), EVs play a crucial role in maternal–embryo interaction by promoting implantation defects which often lead to several pregnancy-related disorders. Maternal immune macrophage-derived EVs are shown to be endocytosed by placental trophoblasts, resulting in the release of pro-inflammatory cytokines, thereby contributing to the maternal inflammatory responses to protect the fetus [162]. On the other hand, placental trophoblast-derived EVs are loaded with chromosome 19 miRNA cluster (C19MC) which attenuates autophagy-mediated virus replication in non-placental cells, thereby protecting the embryo from viral infections [163]. Delorme-Axford [163], unlike others, employed ultracentrifugation followed by density-gradient centrifugation in their EV preparation which is shown to yield highly purified EVs.

*The role of EVs in inflammatory diseases of the endocrine system*. A few studies indicate the active participation of the EVs in inflammatory responses of the endocrine system. For example, EVs derived from obese adipose tissues and plasma show a significantly lower expression of miR-141-3p, which is associated with glucose intolerance and insulin resistance [164,165]. EVs released into the serum from brown adipocytes contain a significant level of miR-99b, which targets FGF21 in the liver, thereby contributing to metabolic dysfunctions such as glucose intolerance in obesity [166]. Adipose tissue macrophage (ATM)-EVs are shown to be over-expressed with miR-155 under obese conditions, which targets PPARγ in adipocytes, myocytes, and primary hepatocytes, leading to glucose intolerance and insulin resistance [167]. As with most cases, the use of ultracentrifugation to isolate EVs in the mentioned studies raises questions about the presence of protein contaminants in the EV suspension.

*EVs of the lymphatic system in inflammatory diseases*. EVs of the lymphatic system often influence various inflammation-associated diseases. For example, the concentration of EVs derived from the lymph is shown to be well-elevated in atherosclerotic conditions as compared to healthy controls, which is believed to contribute to lymphatic dysfunction and associated-inflammatory disease progression [168]. Pronounced inflammation-induced vascular leakage promotes the egress of platelet-derived EVs into the lymphatic system, which is shown to contribute to the pathogenesis of rheumatoid arthritis (RA) [169]. Figure 6 demonstrates the role of EVs in inflammation-related diseases of the urinary system, reproductive system, endocrine system, and lymphatic system.

Apart from the above-mentioned conditions, inflammation is shown to play a major role in the pathogenesis of diseases, associated with memory T-cells. Inflammation is often controlled by the memory T-cells during repeated exposure to infectious agents. The duration of their existence is significantly enhanced by the telomeres, which are shown to be transferred via the EVs in immunological synapse, as discovered recently by Lanna et al. [170]. The intriguing discovery by the group indicates that the interaction of T-cells with the antigen-presenting cells results in the cleavage of telomeres in the antigen-presenting cells and their subsequent incorporation into the EVs at the immunological synapse. These EVs are positive for recombination factor Rad51, which is readily transferred to the T-cells following EV fusion. Inside the T-cells, Rad51-mediated recombination enables the fusion of EVs-carried telomeres with the T-cells’ chromosome ends, leading to an increase in chromosome length. This further contributes to the protection of T-cells from senescence, ultimately imparting long-lasting immune protection [170].

**Table 5 cells-12-01963-t005:** The role of EVs in various inflammatory diseases.

Disease	EVs Found in	Function	Reference/s
*Neuroinflammatory disease*			
MS	CSF and Plasma	Endothelial- or platelet-EVs from MS patients’ plasma promote BBB leakage, resulting in myeloid- and T-cells’ transmigration into CNS contributing to MS neuropathology	[115,116,117]
AD and PD	CSF and Plasma	Microglia and neuronal-EVs from AD or PD patients transport Aβ, α-synuclein, and tau to the local/distant neurons, leading to neuronal loss	[118,119,120,121]
CJD	Plasma	PrP^Sc^ is selectively packaged into neuronal EVs and EV-mediated transfer of PrP^Sc^ contributes to the pathogenetic spread of CJD	[123]
PML	Serum	JCPyV transfer via the serum EVs of PML patients between the glial cells is infectious and contributes to PML pathogenesis	[124]
CM	Plasma	*Plasmodium*-infected red blood cell-derived EVs are implicated in the pathogenesis of CM, and blocking EV biogenesis shows protection against CM	[126]
Stroke	-	MSC-EVs inhibit pro-inflammatory M1 microglial differentiation, preventing neuroinflammation and brain injury following MCAO	[127]
SCI	-	EVs released from infiltrating macrophages are loaded with NOX2 which targets PTEN in the recipient neurons and promotes PI3K-AKT-driven outgrowth	[128]
TBI	Serum	Microglial EVs transfer miR-124-3p to the neurons and target PDE4B to down-regulate the mTOR pathway leading to inhibition of neuronal inflammation and thus promoting neurite growth	[129]
*Cardiovascular inflammatory diseases*			
Atherosclerosis	Plaque and plasma	EVs from atherogenic plaque, monocytes, and neutrophils trigger the endothelial ICAM-1 expression leading to leukocyte recruitment, adhesion, and trans-endothelial migration via pro-inflammatory signaling mechanisms	[85,130,131]
	Plasma	During plaque maturation stages, platelet-EVs trigger the phagocytosis of ox-LDL by macrophages, and adipose cell-derived EVs stimulate cholesterol efflux by macrophages via pro-inflammatory signaling, both of which lead to the formation of foam cells	[132,133]
	Plasma	During plaque progression, EVs from inflammatory macrophages promote microcalcification	[134,135]
MI and ischemic heart disease	Myocardium	Cardiomyocytes and endothelial-EVs induce the release of pro-inflammatory cytokines and chemokines from infiltrating monocytes, thereby contributing to MI and ischemic heart disease progression	[136]
	Myocardium	Activated macrophage-derived miR-155-enriched EVs are incorporated into cardiac fibroblasts and promote inflammation while suppressing fibroblast proliferation, leading to cardiac rupture	[137]
HF	Plasma	Cardiac fibroblast-derived EVs are enriched with miR27a* and miR-21*, promoting cardiac hypertrophy	[138,139]
	Plasma	Cardiomyocyte-derived EVs promote fibroblast proliferation depending on miR-217 transfer	[140]
Aneurysms	Intraluminal thrombus of aortic aneurysm	Neutrophil-EVs carry proteases ADAM10 and ADAM17 which degrade aortic walls	[141]
	Plasma	Ficolin-3 + platelet-EVs often contribute to the progression of aortic aneurysms	[142]
*Respiratory inflammatory diseases*			
ALI or ARDS	BALF	EVs from alveolar macrophages or alveolar type-I epithelial cells upon infection or sterile stimulation, respectively, trigger pro-inflammatory cytokines and mediators’ release from naïve alveolar macrophages, contributing to the lung inflammation	[143]
COPD		EVs from bronchial epithelial cells are enriched with miR-210, which regulates autophagy functions and myofibroblast differentiation, the dysregulation of which leads to COPD pathogenesis	[144]
PH		miR-143-loaded EVs from PASMCs promote migration and differentiation of PAECs, leading to PH pathogenesis	[145]
ILF	BALF	BALF-EVs, loaded with WNT5A, trigger the proliferation of lung fibroblasts, leading to ILF pathogenesis	[146]
Asthma	Plasma	EVs derived from the plasma of asthma patients are related to epithelial and smooth muscle cell functions	[147]
*Inflammatory diseases of the digestive system*			
NEC	Breast milk	BM-EVs protect IEC against H/R-induced apoptosis and loss of proliferation	[149]
	Amniotic fluid	AFSC-EVs promote intestinal epithelial proliferation and anti-inflammation, leading to epithelial regeneration to help intestinal recovery from NEC	[150]
IBD	Intestinal luminal fluid	TGF-β1+ EVs from IECs under physiological conditions induce T_reg_ and immunosuppressive dendritic cells, leading to the downregulation of IBD severity	[152]
	Intestinal mucosa	miR-223+ EVs from MCs target CLDN8 in the IECs, resulting in the loss of intestinal epithelial tight junctions and increased epithelial permeability	[153]
	Serum	ANXA1+ EVs from injury induced IECs help in the activation of the wound repair process	[154]
*Integumentary inflammatory diseases*			
SLE	Plasma	SLE plasma-EVs promote endothelial release of pro-inflammatory cytokines, endothelial apoptosis, and increased vascular permeability, contributing to secondary tissue leukocyte infiltration	[155]
Psoriasis	Plasma	IFN-α-triggered mast cell-derived cytoplasmic PLA_2_+ EVs promote neo-lipid antigen presentation by CD1a+ cells and their concomitant recognition by CD1a-reactive T-cells, leading to IL22 and IL17A release and skin inflammation	[156]
AD	Plasma	SEVs trigger DMECs to induce the expression of E-selectin, VCAM-1, and ICAM-1 as well as IL-6 release to promote endothelial adhesion and subsequent transmigration of leukocytes, leading to AD progression	[157]
*Musculoskeletal inflammatory diseases*			
OP	BMIF	aBMSCs, under oxidative stress, release miR-183-5p-laden EVs which target Hmox1 in yBMSCs, thereby leading to the inhibition of proliferation and osteogenic differentiation as well as senescence induction of yBMSCs	[158]
OA	-	IL-1β-stimulated SFB-derived EVs promote MMP-13 and ADAMTS-5 expression while inhibiting COL2A1 and ACAN expression in articular chondrocytes, leading to OA pathology	[159]
*Urinary inflammatory diseases*			
AKI	Plasma and Urine	Hypoxia or I/R-induced injured TECs release miR-150-loaded EVs which trigger profibrotic manifestations in renal fibroblasts	[171]
IgAN	Urine	CCL2 mRNA expression in urinary EVs of IgAN is significantly higher as compared to controls, which is correlated with tubular interstitial inflammation and C3 deposition, reflecting renal injury and impaired renal functions	[161]
*Reproductive system inflammatory diseases*			
Pregnancy disorders	Plasma	Maternal macrophage derived EVs induce the release of pro-inflammatory cytokines from placental trophoblasts, contributing to maternal inflammatory responses to protect the fetus	[162]
	Amniotic fluid	Placental trophoblast derived EVs, via the transfer of C19MC, prevent virus replication in non-placental cells, leading to embryonic protection against viral infections	[163]
*Inflammatory diseases of the endocrine system*			
Obesity	Adipose tissue and Plasma	Obese adipose tissue or plasma EVs show a significant down-regulation of miR-141-3p expression which contributes to glucose intolerance and insulin resistance	[164,165]
	Serum	Brown adipocyte-derived miR-99b-laden EVs target FGF21 in the liver, leading to metabolic dysfunctions such as glucose intolerance	[166]
	Serum	ATM-EVs are highly expressed with miR-155, which targets PPARγ in adipocytes, myocytes, and primary hepatocytes, leading to glucose intolerance and insulin resistance	[167]
*EVs of the lymphatic system in inflammatory diseases*			
Atherosclerosis	Lymph	Lymph-derived EVs in atherosclerotic conditions influence lymphatic dysfunction and associated inflammatory disease progression	[168]
RA	Lymph	In RA, prolonged inflammation-induced vascular leakage promotes the egress of platelet-derived EVs in the lymphatic system, contributing to the pathogenesis of RA	[169]

*Abbreviations:* MS, multiple sclerosis; CSF, cerebrospinal fluid; BBB, blood–brain barrier; CNS, central nervous system; AD, Alzheimer’s disease; PD, Parkinson’s disease; Aβ, β-amyloid; CJD, Creutzfeldt–Jakob disease; PrP^Sc^, transmissible prion proteins; PML, progressive multifocal leukoencephalopathy; JCPyV, JC polyomavirus; CM, cerebral malaria; MSC, mesenchymal stem cell; MCAO, middle cerebral artery occlusion; SCI, spinal cord injury; NOX2, NADPH oxidase 2; PTEN, phosphatase and tensin homolog; PI3K, phosphatidylinositol 3-kinase; TBI, traumatic brain injury; PDE4B, phosphodiesterase 4B; mTOR, the mammalian target of rapamycin; ICAM-1, intercellular adhesion molecule 1; ox-LDL, oxidized low-density lipoprotein; MI, myocardial infarction; HF, heart failure; ADAM, a disintegrin and metalloproteinase; ALI, acute lung injury; ARDS, acute respiratory distress syndrome; BALF, bronchoalveolar lavage fluid; COPD, chronic obstructive pulmonary disease; PH, pulmonary hypertension; PASMC, pulmonary arterial smooth muscle cell; PAEC, pulmonary arterial endothelial cell; ILF, idiopathic lung fibrosis; WNT5A, Wnt family member 5A; NEC, necrotizing enterocolitis; BM, bone marrow; IEC, intestinal epithelial cell; H/R, hypoxia/reoxygenation; AFSC, amniotic fluid stem cell; IBD, inflammatory bowel disease; TGF-β1, transforming growth factor β1; T_reg_ cell, regulatory T-cell; MC, mast cell; CLDN8, claudin 8; ANXA1, annexin A1; SLE, systemic lupus erythematosus; IFN, interferon; PLA_2_, phospholipase A_2_; CD, cluster of differentiation; IL, interleukin; AD, atopic dermatitis; SEV, *Staphylococcus aureus*-derived EV; DMEC, dermal microvascular endothelial cell; VCAM-1, vascular cell adhesion molecule 1; OP, osteoporosis; BMIF, bone marrow interstitial fluid; aBMSC, aged BM stromal cell; yBMSC, young BMSC; Hmox1, heme oxygenase-1; OA, osteoarthritis; SFB, synovial fibroblast; MMP, matrix metalloproteinase; ADAMTS-5, ADAM metalloproteinase with thrombospondin motifs 5; COL2A1, collagen type II α1; ACAN, aggrecan; AKI, acute kidney injury; I/R, ischemia-reperfusion; TEC, tubular epithelial cell; IgAN, IgA neuropathy; CCL2, (C-C motif) ligand 2; C3, complement component 3; C19MC, chromosome 19 miRNA cluster; FGF21, fibroblast growth factor 21; ATM, adipocyte tissue macrophage; PPARγ, peroxisome proliferator activated receptor γ; RA, rheumatoid arthritis.

The role of EVs in coagulation-associated inflammatory diseases. Blood coagulation is a tightly regulated biological process which prevents excessive bleeding when a blood vessel is injured [172]. Blood coagulation and inflammation are intrinsically related; the activation of one process often leads to the activation of the other [111,112,113]. Vessel injury results in the outburst of thrombin, the central key molecule of the coagulation system, which acts on vascular endothelium to induce the release of pro-inflammatory cytokines [173,174]. Inflammation, on the other hand, often leads to endothelial barrier leakage which further enhances the coagulation process [175,176,177]. A recent study delineates the active involvement of Grb2-associated binder 2 (Gab2) in IL-1β-induced exocytosis of P-selectin and von Willebrand factor (vWF) as well as expression of tissue factor (TF) and VCAM-1, which together often results in the pro-coagulant functions [178]. In the past two decades, EVs have slowly emerged as a key molecule which not only influence the coagulation process but also influence both pro- and anti-inflammatory responses. In most of the cases, the EVs are believed to enhance the coagulation process, due to the presence of pro-coagulant protein TF [179] and negatively charged phospholipid PS [180] on the surface. Although EVs’ TF directly activates the coagulation cascade, PS-dependent activation of the coagulation system requires the assembly of factor VIIIa, IXa, and X (tenase complex) as well as factors Va, Xa, and thrombin (prothrombinase complex) in the presence of Ca^2+^ [181]. In contrast to the above, EVs also exert anticoagulant properties. For example, EVs released from the endothelial cells upon exposure with anticoagulant protease activated protein C (APC) turn out to be anti-coagulant [182]; however, in this case, the anticoagulant activity is largely due to the bound APC on the EVs’ surface [182]. Similar to coagulation, EVs also confer both pro- and anti-inflammatory responses in the context of clotting. The pro-inflammatory effects of the platelet-derived EVs are well-established, which prevent their clinical use as a pro-coagulant factor against hypo-coagulable conditions, such as trauma-induced coagulopathy (TIC) [183,184]. On the other hand, Njock et al. demonstrated that EVs released from unperturbed endothelium confer anti-inflammatory responses via the enrichment of anti-inflammatory miRNAs [185]. Despite the advancement of EV research, it was still unknown how EVs generated from the unperturbed vascular endothelium upon exposure of coagulation proteases contribute to the inflammatory responses, until the recent intriguing discovery by Das et al. who demonstrated, for the first time, that FVIIa-triggered endothelial EVs (EEVs) suppress monocytic inflammation against bacterial-LPS-induced sepsis (Figure 7) [8,19]. The study delineates the fact that FVIIa triggers the endothelial release of EVs by endothelial cell protein C receptor (EPCR)-driven activation of protease activated receptor 1 (PAR1) both in vitro and in vivo [19]. Unlike FVIIa-TF-PAR2 signaling, observed predominantly in cancer [186], FVIIa-induced EV generation from unperturbed endothelial cells is shown to be independent of both TF and PAR2 [19]. FVIIa-EEVs are enriched with anti-inflammatory miRNAs, the predominant being miR-10a, and the transfer of EVs-miR-10a to monocytes confers anti-inflammatory responses against LPS-induced sepsis [8,19]. Furthermore, FVIIa infusion into hemophilia patients increases the level of plasma EEVs enriched with miR-10a [4], and these EEVs also impart miR-10a-dependent anti-inflammatory responses [4].

EVs in inflammation therapy. EVs are known for transporting bioactive cargoes, such as proteins, nucleic acids, lipids, etc., between the cells, thereby playing an important role in cell–cell communication. EVs could be engineered at the surface and bestowed with target-specific moiety, rendering the target-specific therapeutic applications of the EVs in various inflammation-associated diseases. For example, therapeutic drugs entrapped within the EVs often reach the target-specific sites with higher efficacy through EVs. The present section provides a brief overview of how EVs could be used as a potential therapeutic agent in the context of various inflammatory diseases (Table 6).

*Therapeutic roles of EVs in neuroinflammatory diseases*. MSC-derived EVs are often used as a promising therapeutic mode in various neuroinflammatory diseases. For example, the activation of infiltrating leukocytes, astrocytes, and microglial cells has been shown to be attenuated upon intra-arterial injection of MSC-EVs in an ischemic stroke-induced rat model [187]. Moreover, T-lymphoblast-derived EVs, packaged with a neuroprotective drug, curcumin, are efficiently taken up by the inflamed brain microglial cells, thereby triggering apoptosis, when administered intranasally in a LPS-induced brain inflammation murine model [188]. Furthermore, in a cocaine-induced brain inflammation murine model, DC-secreted EVs, further engineered to over-express miR-124, are shown to attenuate microglial activation and the expression of pro-inflammatory molecules, TLR4, MYD88, STAT3, and NF-ĸB p65 [189].

*Therapeutic roles of EVs in cardiovascular inflammatory diseases*. EVs often exert beneficial roles in cardiovascular inflammatory diseases in the context of post-MI cardiac repair processes. For example, in an I/R-induced cardiac inflammation rat model, cardiosphere-derived cell (CDC)-derived EVs, laden with Y-RNA fragments, promote the release of IL-10 in the infarcted myocardium, thereby contributing to post-MI cardiac repair [190]. Moreover, DC-EVs are shown to activate CD4 + T-cells, leading to the perturbation of pro-inflammatory cytokines’ release and improvement of cardiac functions in a cardiac MI mice model [191]. Again, adipose-derived stem cell (ADSC)-derived EVs are found to over-express miR-93-5p, which down-regulates autophagy and pro-inflammation by targeting Atg7 and TLR4, respectively, thereby showing protection against infarction-induced myocardial damage in an ischemia-induced cardiac injury rat model [192].

*EVs’ therapeutic roles in respiratory inflammatory diseases*. MSC-EVs often show promising effects against respiratory inflammatory diseases, which render them to be considered an effective therapeutic entity. For example, MSC-EVs, via delivering miR-21-5p, protect the epithelial cells from reactive oxygen species (ROS)-induced apoptotic damage in asthma and COPD [193,194]. Moreover, MSC-EVs also trigger the polarization of alveolar macrophages into M2 phenotypes, thereby inducing the release of anti-inflammatory cytokines and promoting wound healing [195]. MSC-EVs also produce promising therapeutic outcomes in ALI/ARDS by inhibiting the proliferation and differentiation of B-cells as well as promoting the differentiation of T_H_-cells to T_reg_ cells, leading to the down-regulation of pro-inflammatory cytokines TNF-α, IL-1β, and IFN-γ and up-regulation of anti-inflammatory cytokines PEG_2_, IL-10, and TGF-β [196].

*Therapeutic roles of EVs in integumentary inflammatory diseases*. Research indicates that EVs also confer therapeutic potential in several integumentary inflammatory diseases. For example, human keratinocyte (HK)-derived EVs are shown to carry miR-21, which not only promotes angiogenesis but also facilitates fibroblast functions, leading to skin wound healing in diabetic rats [197]. Murine intraperitoneal injection of the EVs, engineered with a super repressor of IĸB (srIĸB, the dominant active form) by the optogenetic method, are shown to inhibit the NF-ĸB pro-inflammatory signaling pathway in liver and spleen neutrophils and monocytes, leading to the attenuation of sepsis-induced inflammation and associated mortality [198].

*EVs’ therapeutic roles in autoimmune inflammatory diseases*. EVs also play their therapeutic roles in autoimmune inflammatory disorders. For example, EVs from IL-10-treated DCs are associated with the inhibition of arthritis onset as well as the already-established severity [199]. Furthermore, in a collagen-induced arthritis (CIA) model of RA, MSC-EVs exert anti-inflammatory effects on B- and T-lymphocytes, thus demonstrating the therapeutic potential of MSC-EVs in RA [200].

Due to commendable success in preclinical studies, EVs have now mostly reached phase I and phase II clinical trials as briefly mentioned in Table 7 [201].

Artificial EVs and disease therapy. So far, the present review discussed how EVs influence different inflammation-associated diseases and their potential use in therapeutic purposes. The natural tropism [202], fine biodistribution and less clearance from the system [203], ability to transfer bioactive cargoes efficiently [39], biocompatibility [204], and most importantly, the extraordinary capacity to cross blood–brain barriers (BBB) [205] render EVs to be an excellent means in various disease therapies. However, the natural EV-based therapies also have their limitations: (1) the heterogeneity of EVs makes EV isolation and purification difficult [206]; (2) although less, EVs still show immunogenic responses [207]; and (3) as this review already discussed, most of the conventional methods of EV isolation are time-consuming, with the yield and purity always remaining a concern. However, in the past decade, the concept of bioengineered EVs has evolved, which includes the isolation of natural EVs followed by some modifications to generate the biomimetic nanocarriers which are not only being used as an efficient drug delivery system but also improved the target specificity significantly [208,209,210]. Recently, two unique mechanisms have evolved. The first one, termed as top-down method, employs the disruption of membranes into small fragments which reassembles automatically to form nano- or microvesicles [211,212]. In the second approach, molecular components such as synthetic lipids are used to generate the artificial lipid bilayers which mimic the EVs [213]. These recently developed artificial EVs have several advantages over the natural EVs [214] which include the fact that (1) the size of the EVs can be easily controlled, and in this sense these EVs reduce the heterogeneity unlike natural EVs, (2) the ingredients, such as synthetic lipids, are commercially available, (3) more standardized and high scale production can be achieved, and (4) these EVs are safe to be used and are highly reproducible. However, additional bioengineering on these artificial EVs could improve the target specificity which is essential in the delivery of EV-based therapeutic drugs against various diseases, including inflammation-associated disorders.

## 2. Conclusions and Future Directions

There is much evidence suggesting that submicron-sized EVs are crucial and have an immense therapeutic approach because of their biocompatibility, at the experimental and clinical level, against several inflammation-associated diseases such as neurological disorders, cardiovascular anomalies, respiratory syndromes, integumentary disorders, and autoimmune diseases and in regenerative medicine. EVs play a vital role in innate and adaptive immunity, including inflammation, as they carry autoantigens, cytokines, lipid mediators, tissue-degrading enzymes, etc. As a non-cellular membrane structure, EVs can regulate immune cell activity, suppress excessive inflammation, and promote immune tolerance. This suggests that EV-based therapies may hold promise for the treatment of inflammatory diseases, providing a targeted and personalized approach. EVs are safe, are efficient, and have many advantages in clinical application due to low immunogenicity, flexibility in engineering the membrane or cargo, and potential for tissue-specific targeting, long half-life, in vivo stability, and high delivery efficiency. Now, several EV-based treatments are being studied in phase I and II clinical trials. EVs’ research is rapidly advancing, and several directions are being explored in the future. Although EVs hold great promise for clinical applications, the toxicity, long-term safety, and immunoregulatory functions of EVs in the human body need to be further evaluated. Additionally, there are several challenges to overcome, including standardization of isolation and characterization, cost-effective production methods with consistent quality, developing robust cargo loading strategies and improving cargo stability within the EVs and biodistribution, targeted delivery and tissue specificity, and establishing clear regulatory guidelines and frameworks to ensure safety, quality, and efficacy for gaining regulatory approvals. To address these challenges, researchers, clinicians, regulators, and industry partners will have to collaborate. In order to overcome these hurdles and unlock the full therapeutic potential of EVs in clinical settings, further research and clinical trials are required to fully explore the therapeutic efficacy and safety of EV-based therapies. Nevertheless, the potential impact of EVs in revolutionizing disease treatment is undeniable. However, if the aforementioned difficulties are addressed, EVs could be used as an excellent mode of delivery system in the therapeutic implications of various inflammation-associated diseases. In addition to these, the present review also delineates how EVs play their part in promoting inflammation-associated diseases. In these conditions, the therapeutic approach includes targeting the EVs. Numerous studies indicate that intracellular signaling mechanisms, leading to the actomyosin reorganization, play a critical role in the biogenesis of EVs. Therefore, targeting EVs’ formation could be a potential therapeutic means to limit EV-associated promotion of inflammatory diseases.

## Figures and Tables

**Figure 1 cells-12-01963-f001:**
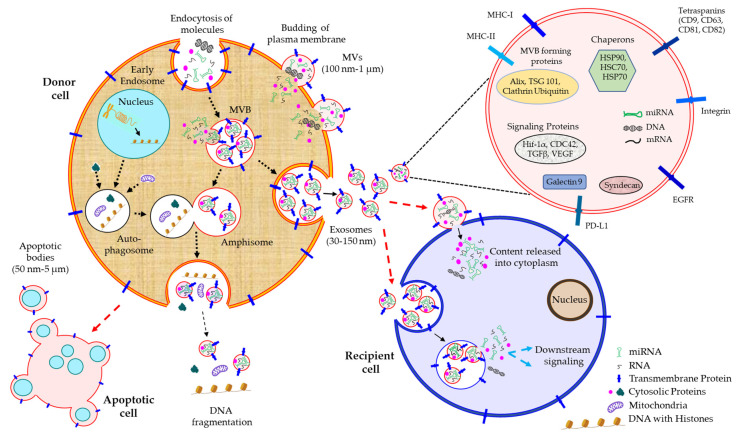
Biogenesis and uptake of EVs. EVs are composed of microvesicles (MVs), exosomes, and apoptotic bodies. MVs are produced by outward budding of the plasma membrane, whereas exosomes are generated by endocytic mechanism. Invagination of the early endosomal membrane produces the exosomes inside the endocytic vesicles which mature into multivesicular bodies (MVBs). MVBs are eventually fused directly with the plasma membrane to release the exosomes outside the cells. Sometimes, MVBs also fuse with the autophagosomes to form amphisomes. Amphisomes, in turn, fuse with the plasma membrane to release their content including the exosomes outside the cells. Apoptotic bodies, on the other hand, are generated during the contraction of the cells, leading to the dissociation of plasma membrane from the cytoskeleton. The induction of apoptosis often results in the fragmentation of DNA which is incorporated into the apoptotic bodies. Both MVs and exosomes, which carry cargoes in the form of RNA, miRNA, proteins, etc., are readily taken up by the recipient cells via either direct fusion with the plasma membrane or endocytosis. In the case of endocytosis, inside the recipient cells, the EVs are further fused with the membrane of endocytic vesicles, thereby releasing the cargoes into the recipient cells’ cytosol. In contrast, direct fusion of EVs with the target cells’ plasma membrane results in the release of the EVs’ cargoes in the cytosol of the recipient cells.

**Figure 2 cells-12-01963-f002:**
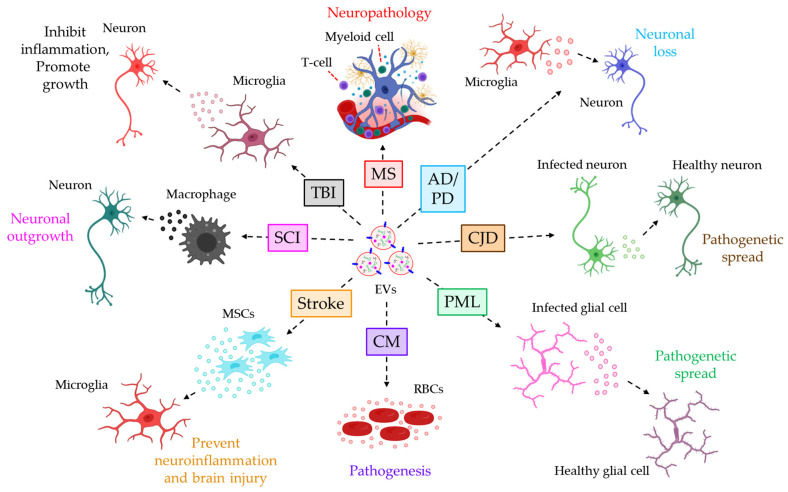
The role of EVs in different forms of neuroinflammatory diseases. EVs from multiple sclerosis (MS) patients induce BBB permeability, leading to the transmigration of T-cells and myeloid cells into the CNS, contributing to MS neuropathology. In the case of Alzheimer’s disease (AD) and Parkinson’s disease (PD), microglial EVs, enriched with neurotoxic molecules, are incorporated into the neurons, leading to neuronal loss. EVs, loaded with infectious PrP^Sc^, are released from infected neurons and are readily incorporated into healthy neurons, leading to pathogenic spread of Creutzfeldt–Jakob disease (CJD). In progressive multifocal leukoencephalopathy (PML), JCPyV-laden EVs are transferred between the glial cells which contribute to the pathogenesis of PML. *Plasmodium*-infected RBC-derived EVs are also known for increasing the pathogenesis of cerebral malaria (CM). In stroke, MSCs-EVs perturb microglial neuroinflammatory responses and the subsequent brain injury. Again, in spinal cord injury (SCI), EVs from infiltrating macrophages transport NOX2 to the neuronal cells which leads to the regeneration of neuronal outgrowth. Similarly, microglial EVs carried miR-124-3p, which not only prohibits neuronal inflammation but also induces neurite growth in the context of traumatic brain injury (TBI).

**Figure 3 cells-12-01963-f003:**
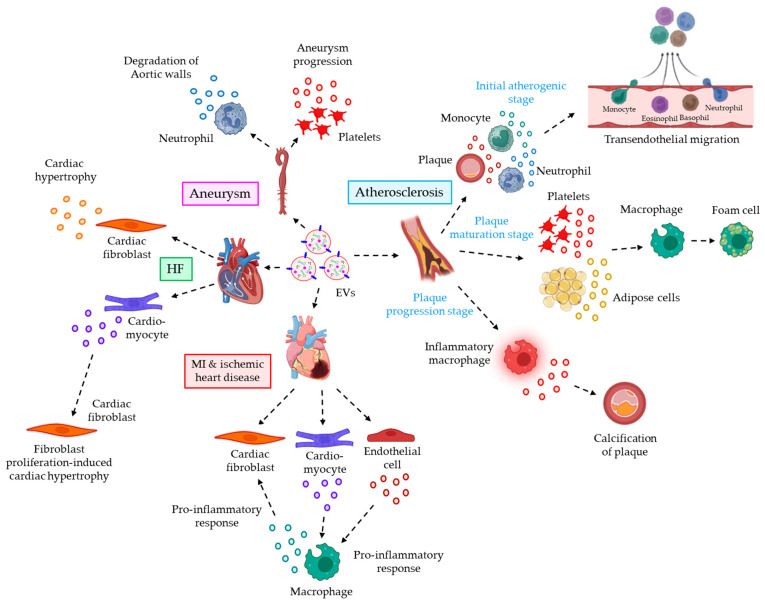
The role of EVs in cardiovascular inflammatory diseases. EVs are shown to play important roles in the progression of various cardiovascular inflammatory diseases including atherosclerosis, myocardial infarction (MI) and ischemic heart disease, heart failure (HF), and aneurysm. In atherosclerosis, during initial atherogenic stages, monocyte-, neutrophil-, and plaque-derived EVs interact with the endothelium, leading to transendothelial migration of leukocytes. During plaque maturation stages, platelet- and adipose cell-derived EVs convert macrophages into foam cells. Furthermore, during plaque progression stages, inflammatory macrophage derived EVs promote calcification of the plaque. In MI and ischemic heart disease, endothelial cell- and cardiomyocyte-derived EVs trigger macrophage pro-inflammatory responses. On the other hand, macrophage-EVs induce cardiac fibroblasts’ pro-inflammatory responses. In the case of HF, cardiomyocyte derived EVs promote the proliferation of cardiac fibroblasts, thereby contributing to cardiac hypertrophy. Moreover, EVs generated from cardiac fibroblasts also trigger cardiac hypertrophy. In aortic aneurysms, neutrophil-derived EVs cause degradation of aortic walls whereas platelet-derived EVs contribute to the progression of aneurysms.

**Figure 4 cells-12-01963-f004:**
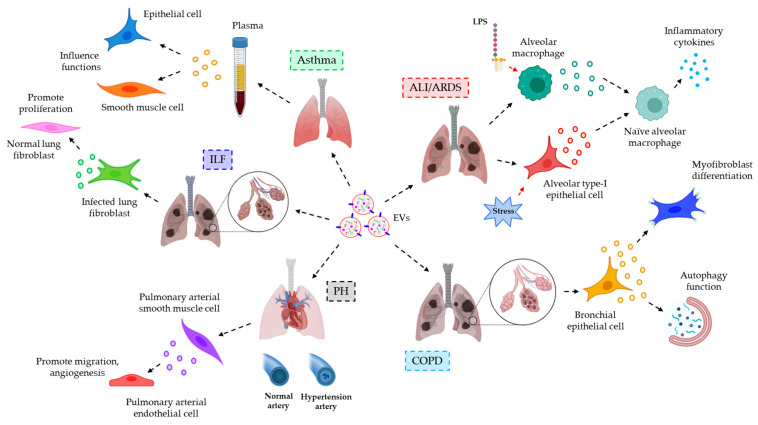
The role of EVs in various respiratory inflammatory diseases. EVs’ role is well-established in various inflammation-associated respiratory diseases, such as acute lung injury (ALI) or acute respiratory distress syndrome (ARDS), chronic obstructive pulmonary disease (COPD), pulmonary hypertension (PH), idiopathic lung fibrosis (ILF), and asthma. In ALI/ARDS, infected alveolar macrophage- and oxidative stress-induced alveolar type-I epithelial cell-derived EVs trigger pro-inflammatory cytokines’ release from naïve alveolar macrophages. In COPD, bronchial epithelial cell secreted EVs promote myofibroblast differentiation and influence autophagy functions. In PH, PASMC-released EVs promote migration and angiogenesis of PAECs. ILF-infected lung fibroblast-derived EVs promote proliferation of normal lung fibroblasts. EVs from the plasma of asthma patients influence the functions of lung epithelial and smooth muscle cells.

**Figure 5 cells-12-01963-f005:**
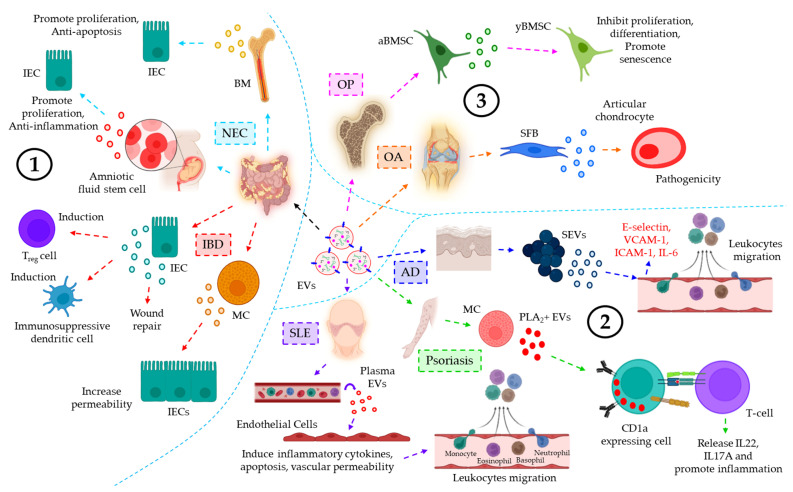
The role of EVs in various inflammatory diseases associated with the digestive system, integumentary system, and musculoskeletal system. (**1**) The role of EVs in digestive inflammatory diseases. In necrotizing enterocolitis (NEC) (sky dotted arrows), BM-EVs promote IEC proliferation and anti-apoptosis. Moreover, amniotic fluid stem cell derived EVs promote the proliferation and anti-inflammation of IEC. In inflammatory bowel disease (IBD) (red dotted arrows), IEC-EVs promote the induction of T_reg_ and immunosuppressive dendritic cells, as well as wound repair. MC-derived EVs increase IEC permeability. (**2**) EVs’ roles in integumentary inflammatory diseases. In systemic lupus erythematosus (SLE) (violet dotted arrows), plasma EVs promote endothelial apoptosis, permeability, and release of pro-inflammatory cytokines, leading to leukocytes transmigration. In psoriasis (green dotted arrows), MC-EVs containing PLA_2_ are taken up by CD1a-expressing cells which present a lipid antigen (red dot) to the CD1a-reactive T-cell, leading to the release of pro-inflammatory cytokines IL22 and IL17A. In atopic dermatitis (AD) (blue dotted arrows), SEVs trigger the expression of E-selectin, VCAM-1, and ICAM-1 and the release of IL-6, thereby promoting vascular permeability to induce leukocytes transendothelial migration. (**3**) EVs in musculoskeletal diseases. In osteoporosis (OP) (pink dotted arrows), aBMSC-derived EVs inhibit proliferation and differentiation, while promoting senescence of yBMSC. In the case of osteoarthritis (OA) (orange dotted arrows), SFB-EVs induce pathogenicity to articular chondrocytes.

**Figure 6 cells-12-01963-f006:**
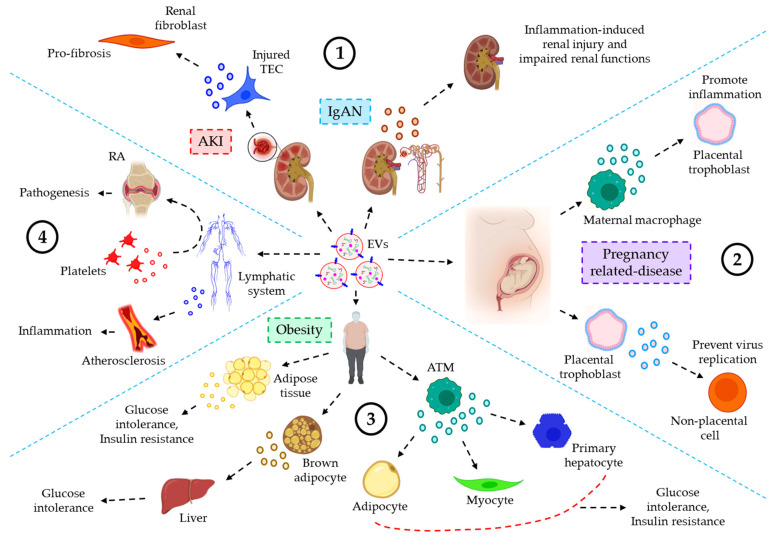
The role of EVs in various inflammatory diseases associated with the urinary, reproductive, endocrine, and lymphatic system. (**1**) EVs and urinary inflammatory diseases. In acute kidney injury (AKI), I/R-induced TEC-derived miR-150-laden EVs promote profibrotic manifestations to renal fibroblasts. In IgA nephropathy (IgAN), CCL2 mRNA-loaded EVs promote inflammation-induced renal injury and impaired renal functions. (**2**) EVs’ roles in reproductive inflammatory diseases. In pregnancy-related diseases, maternal macrophage-derived EVs trigger inflammatory responses in the placental trophoblast. On the other hand, EVs from placental trophoblasts prevent virus replication of non-placental cells, thereby protecting the embryo from viral infections. (**3**) EVs and endocrine inflammatory responses. In obesity, adipose tissue derived EVs influence glucose intolerance and insulin resistance. Again, brown adipocyte derived EVs promote glucose intolerance after migrating to the liver tissues. Furthermore, ATM-EVs target adipocytes, myocytes, and primary hepatocytes leading to glucose intolerance and insulin resistance. (**4**) EVs of the lymphatic system influencing various inflammatory diseases. In atherosclerosis, EVs’ level in the lymph is significantly increased, contributing to inflammation and associated disease progression. On the other hand, in rheumatoid arthritis (RA), inflammation-induced vascular leakage renders the transmigration of platelet-EVs into the lymph, thereby contributing to the pathogenesis of RA.

**Figure 7 cells-12-01963-f007:**
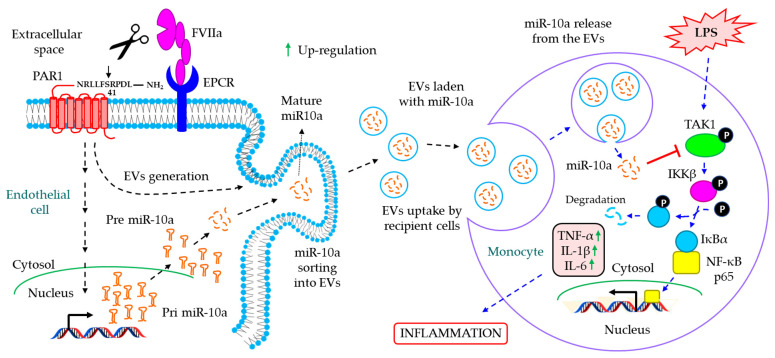
Schematic representation showing the release of miR-10a-enriched EVs from endothelial cells and their uptake by the recipient monocytes, leading to the phenotypic alteration. FVIIa binding to the endothelial EPCR leads to the cleavage of PAR1 at R41 which not only triggers the induction of miR-10a expression but also promotes the release of miR-10a-laden EVs. The FVIIa-released EVs are readily taken up by the recipient monocytes via endocytic mechanism, ultimately resulting in the release of EVs’ miR-10a into the recipient cell’s cytosol. The released cytosolic miR-10a targets TAK1, thereby preventing LPS-induced activation of the NF-ĸB pathway to increase the release of TNF-α, IL-1β, and IL-6 and subsequent pro-inflammatory responses.

**Table 1 cells-12-01963-t001:** The size, marker, and biogenetic mechanism of different forms of EVs.

EVs’ Type	Size (Diameter)	Marker/s	Biogenetic Mechanism
Microvesicles	100 nm to 1 µm	Tetraspanins	Microvesicles are generated by outward budding of the plasma membrane of the cell
Exosomes	30 nm to 150 nm	TSG101, Alix, HSP90β, HSC70	Invagination of the early endosomal membrane produces exosomes which mature into multivesicular bodies (MVBs). MVBs fuse with the plasma membrane to release the exosomes outside the cells. MVBs can also fuse with auto-phagosomes to form amphisomes which eventually fuse with the plasma membrane to release the exosomes from the cells
Apoptotic bodies	50 nm to 5 µm	HSP60, GRP78, Histones	Increased hydrodynamic forces, generated during apoptosis-induced cell contraction, segregate the plasma membrane from cytoskeleton to release such bodies

*Abbreviations:* TSG101, tumor susceptibility gene 101; Alix, ALG-2-interacting protein X; HSP, heat shock protein; HSC70, heat shock cognate protein 70; GRP78, glucose-regulated protein 78.

**Table 3 cells-12-01963-t003:** Purity and recovery of the EVs among different EV isolation techniques according to MISEV2018 [73].

EVs Isolation Technique	Purity	Recovery
Precipitation kits or polymer (PEG or others)	Low	High
Low molecular weight cut off centrifugal filters with no further separation steps	Low	High
High speed ultracentrifugation with no previous lower speed steps	Low	High
Size-exclusion chromatography	Moderate	Moderate
High molecular weight centrifugal filters	Moderate	Moderate
Differential ultracentrifugation with intermediate time/speed with/without wash	Moderate	Moderate
Tangential flow filtration	Moderate	Moderate
Membrane affinity columns	Moderate	Moderate
Filtration combined with size-exclusion chromatography	High	Low
Immuno- or other affinity isolation with flow cytometry	High	Low
Surface charge-based isolation techniques	High	Low

*Note:* High purity with high recovery is difficult to achieve as per MISEV2018 [73].

**Table 6 cells-12-01963-t006:** Therapeutic roles of EVs in various inflammatory diseases.

Disease	Model	EVs’ Source	Function	Reference
Neuroinflammatory diseases	Ischemic stroke	MSC	MSC-EVs prevent activation of astrocytes, infiltrating leukocytes, and microglial cells	[187]
	LPS-induced brain inflammation	T-lymphoblast	Curcumin-laden EVs induce apoptosis of inflamed brain microglial cells	[188]
	Cocaine-induced brain inflammation	DC	miR-124-laden EVs attenuate microglial activation and expression of pro-inflammatory mediators, TLR4, MYD88, STAT3, and NF-ĸB p65	[189]
Cardiovascular inflammatory diseases	I/R-induced cardiac	CDC	CDC-EVs with Y-RNA fragments promote IL-10 release in the infarcted myocardium and trigger inflammation post-MI cardiac repair	[190]
	Cardiac MI inflammation	DC	DC-EVs promote IL-10 release from CD4+T-cells, reducing inflammation and improved cardiac functions	[191]
	Ischemia-induced cardiac injury	ADSC	ADSC-EVs with miR-93-5p target Atg7 and TLR4, thereby attenuating autophagy and inflammation to protect against infarction-induced myocardial damage	[192]
Respiratory inflammatory diseases	Asthma and COPD	MSC	MSC-EVs’ miR-21-5p targets ROS-triggered apoptotic pathway in epithelial cells	[193,194]
	Lung inflammation	MSC	MSC-EVs promote the conversion of alveolar macrophages into M2 phenotypes, leading to anti-inflammation and would healing	[195]
	ALI/ARDS	MSC	MSC-EVs inhibit proliferation and differentiation of B-cells and promote differentiation of T_H_-cells to T_reg_ cells, leading to anti-inflammatory cytokines release while attenuating pro-inflammatory cytokines	[196]
Integumentary inflammatory disease	Diabetes	HK	miR-21+ HK-EVs promote angiogenesis and facilitate fibroblast function, leading to skin wound healing	[197]
	Sepsis-induced inflammation	-	srIĸB-EVs inhibit the NF-ĸB pathway in neutrophils and monocytes, alleviating sepsis-induced inflammatory responses	[198]
Autoimmune inflammatory diseases	Arthritis	DC	IL-10-treated DC-EVs not only inhibit the onset of arthritis but also lower the severity of already-established disease	[199]
	Collagen-induced RA	MSC	MSC-EVs exert anti-inflammatory effects on B- and T-lymphocytes	[200]

*Abbreviations*: MSC, mesenchymal stem cell; LPS, lipopolysaccharide; DC, dendritic cell; TLR4, Toll-like receptor 4; MYD88, myeloid differentiation primary response 88; STAT3, signal transducer and activator of transcription 3; NF-ĸB, nuclear factor kappa-light-chain-enhancer of activated B-cells; I/R, ischemic-reperfusion; CDC, cardiosphere-derived cell; IL, interleukin; MI, myocardial infarction; CD, cluster of differentiation; ADSC, adipose-derived stem cell; Atg7, autophagy related 7; COPD, chronic obstructive pulmonary disease; ROS, reactive oxygen species; ALI, acute lung injury; ARDS, acute respiratory distress syndrome; T_H_, helper T; T_reg_, regulatory T; HK, human keratinocyte; srIĸB, super repressor of IĸB; RA, rheumatoid arthritis.

**Table 7 cells-12-01963-t007:** EVs, associated with inflammatory diseases in human clinical trials.

NTA Number	Disease	Phase	EVs’ Source	Age and Sex	No. of Participants	Recruitment Status
NCT03384433	Cerebrovascular disease	I and II	Allogenic MSCs	40–80 years, both M and F	5	Unknown by 15 January 2021
NCT04602104	ARDS	I and II	Allogenic human MSCs	18–70 years, both M and F	169	Unknown by 2 November 2021
NCT04493242	COVID-19, ARDS	II	BM-MSCs	18–85 years	102	Completed by 11 April 2023
NCT04602442	SARS-CoV-2 pneumonia	II	MSCs	18–65 years both M and F	90	Unknown by 26 October 2020
NCT04276987	Coronavirus	I	Allogenic adipose MSCs	18–75 years both M and F	24	Completed by 7 September 2020
NCT02565264	Ulcer	Early I	Platelets	Child, adult, Older adult	5	Unknown by 9 September 2020
NCT04664738	Skin graft	I	Platelets	18–75 years both M and F	37	Enrolling by invitation by 27 June 2023
NCT02138331	Diabetes Mellitus type 1	II and III	MSCs	18–60 years both M and F	20	Unknown by 14 May 2014

*Abbreviations:* MSC, mesenchymal stem cell; ARDS, acute respiratory distress syndrome; COVID-19, Coronavirus disease 2019; SARS-CoV-2, severe acute respiratory syndrome coronavirus 2; BM, bone marrow; M, male; F, female; *Note:* Recruitment status; ‘Unknown’, A study on Clinical Trials.gov whose last known status was recruiting; not yet recruiting; or active, not recruiting but that has passed its completion date, and the status has not been last verified within the past 2 years; ‘Completed’, The study has ended normally, and participants are no longer being examined or treated (that is, the last participant’s last visit has occurred); ‘Enrolling by invitation’, The study is selecting its participants from a population, or group of people, decided on by the researchers in advance. These studies are not open to everyone who meets the eligibility criteria but only to people in that particular population, who are specifically invited to participate.

## Data Availability

Not applicable.

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
