# Peer review of "Beyond Macromolecules: Extracellular Vesicles as Regulators of Inflammatory Diseases"

_cells, 2023, doi:10.3390/cells12151963_

Round 1

Reviewer 1 Report

The article is deal with the in extracellular vesicles as regulators of inflammatory diseases. The topic discussed is very important for the identification of mechanisms regulating inflammation.

I would like to make some comments:

In Figures 1, 2 it is necessary to provide a transcript of the abbreviations.

Lines 71, 86: It might be better to replace “Inward invagination” with “invagination”.

Lines 216 and 217: the sentence is disconnected? The same with other figures. Please, display all figures after the end of sentences.

 Inflammation plays a key role in the pathogenesis of various diseases. The control of inflammation during repeated exposure to infectious agents is based on memory T cells. The duration of their existence can be increased due to telomeres, which are transmitted by extracellular vesicles in the immunological synapse. It is necessary to discuss and cite the article:

Lanna, A., Vaz, B., D’Ambra, C. et al. An intercellular transfer of telomeres rescues T cells from senescence and promotes long-term immunological memory. Nat Cell Biol 24, 1461–1474 (2022). https://doi.org/10.1038/s41556-022-00991-z

In the discussion, it is also desirable to consider the possibility of controlling the formation of vesicles as one of the therapeutic approaches.

Reviewer 2 Report

Dear authors, the review describes the role of Evs in inflammation and current therapies. I found it interesting, supported by a robust bibliography. The figures and tables are clear and full of information. I would suggest some small changes before publication. 

Revising the paragraphs "EVs and cardiovascular inflammatory responses" and "EVs in respiratory inflammatory diseases." There are sentences unclear, a little too long and difficult to understand.

Standardize the references;

the title of the table 1 is dirty;

line 76 and 350 have an extra space;

the list of abbreviations would be easier to understand if it were in alphabetical order.

Reviewer 3 Report

It is an illustrative revision about the role of EVs in inflammation with pertinent examples mentioned in each section. It addresses a very broad topic which is an opportunity to be didactic, mentioning the generalities about EVs in lines 39-119; and critic reflecting on the importance of EVs in inflammation therapy in lines 599-626.  However, considering the broadness of the topic, the manuscript must be improved in the following aspects:

11.       As authors are experts in the field, it would be enlightening for readers some comments about the different, and recommended experimental methods to enrich extracellular vesicles. Comparing contaminants (cytokines, lipoproteins, etc.), complexity, yields and time consumption.  

22.       Explicitly recognize the heterogeneity of EVs preparations and the importance of MISEV (the Minimal Information for Studies of Extracellular Vesicles, DOI: 10.1080/20013078.2018.1535750 ) recommendations.

33.       Discuss on the possibility that the preparations of enriched EVs used in the cited references, are contaminated with soluble extracellular cytokines and that the observed effects on inflammation could be the result of both, EVs and contaminant cytokines.

44.       Please briefly comment on What is known about selectivity of target cells? Among the wide variety of circulating EVs in blood, Which molecules determine the target cells of a specific EVs subpopulation? Provide examples of protein interactions that determine selectivity.  

55.       In lines 498-523 there is a detailed description of the author’s own research (references 4, 8, 10 and 17). A review must be neutral and the different topics must be described in a balanced way. Please make this description much shorter and refer to Figure 7 as an “illustrative example of the crosstalk between inflammation and coagulation”.

66.       As authors are experts in the field, it would be enlightening for readers some comments about the possibility that EVs were artificially produced mixing bioactive phospholipids, proteins and RNAs to avoid EVs purification from cell cultures, as an alternative for production of therapeutic EVs.

77.       Lines 602-603. As the therapeutic use of EVs is an important part of the manuscript, it would be enlightening for readers some description of the EVs-based clinical trials. Please add a table with information of such trials.

88.       The Conclusion section is mainly about the therapeutic potential of EVs. However, the title and most of the text is about EVs and inflammation. Please add appropriate conclusions about EVs role inflammatory diseases.

99.        As authors are experts in the field, it would be enlightening for readers some comments about the main challenges and future directions on this field. Please add some comments on this.  

 Minor points

10.   Please update references 49, 51, 52, 53 and 54. The general aspects of inflammation on damaged cells, recruitment of immune cells, healing processes and chronic disorders, must be available in recent publications instead of reviews from 2000-2016.

111.   Line 41. Secreted instead of secretary.

112.   Line 387. The colon after “implantation” and before “defects” should not be there.  

Round 2

Reviewer 3 Report

My previous comments have been properly addressed.

In the new text, authors mention that EV’s isolation using the ExoQuick kit, without subsequent ultracentrifugation, is associated with high purity (Reference 150 in line 635, ref. 152 in line 585, and ref. 159 in line 585). However, in my experience, the use of precipitation reagents in commercial EV’s enrichment kits, results in high level of contamination with lipoproteins as determined by mass spectrometry analyses of the EV’s pellets. I recommend authors to be more cautious in the comments about the use of ExoQuick as a single enrichment step.  

The word “phase” in Table 7 has two letters “a”.
